# Opinion: Eliminating aircraft soot emissions

Una Trivanovic and Sotiris E. Pratsinis

Particle Technology Laboratory, Department of Mechanical and Process Engineering, ETH Zürich, 8092, Switzerland

*Correspondence to:* Sotiris E. Pratsinis (sotiris.pratsinis@ptl.mavt.ethz.ch)

## 1 Abstract

Soot from aircraft engines deteriorates air quality around airports and can contribute to climate change primarily by influencing cloud processes and contrail formation. Simultaneously, aircraft engines emit carbon dioxide ($CO_2$). nitrogen oxides (NOx) and other pollutants which also negatively affect human health and the environment. While urgent action is needed to reduce all pollutants, strategies to reduce one pollutant may increase another, calling for a need to decrease, for example, the uncertainty associated with soot's contribution to net Radiative Forcing (RF) in order to design targeted policies that minimize the formation and release of all pollutants. Aircraft soot is characterized by rather small median mobility diameters, $d_m = 8 – 60$ nm, and at high thrust, low (< 25%) organic carbon to total carbon (OC/TC) ratios while at low thrust the OC/TC can be quite high (> 75%). Computational models could aid in the design of new aircraft combustors to reduce emissions, but current models struggle to capture the soot $d_m$, and volume fraction, $f_v$ measured experimentally. This may be in part due to oversimplification of soot's irregular morphology in models and a still poor understanding of soot inception. Nonetheless, combustor design can significantly reduce soot emissions through extensive oxidation or near-premixed, lean combustion. For example, lean premixed prevaporized combustors significantly reduce emissions at high thrust by allowing injected fuel to fully vaporize before ignition while low temperatures from very lean jet fuel combustion limit the formation of NOx. Alternative fuels can be used alongside improved combustor technologies to reduce soot emissions. However, current policies and low supply promote the blending of alternative fuels at low ratios (~1%) for all flights, rather than using high ratios (> 30%) in a few flights which could meaningfully reduce soot emissions. Here, existing technologies for reducing such emissions through combustor and fuel design will be reviewed to identify strategies that eliminate them.

## 1. Introduction

Aviation is a growing industry with a significant impact on human health and the environment due to the emission of combustion by-products, including soot aerosols. The latter is one of the most important contributors to climate change (Bond et al., 2013) and a component of air pollution known to cause cancer, cardiovascular and respiratory diseases, and it has been correlated with various other illnesses (Niranjan and Thakur, 2017). For aviation in particular, the adverse health effects of aircraft emissions are partly due to non-volatile particle matter (Delaval et al., 2022) and aircraft soot has similar toxicity to diesel exhaust particles (Bendtsen et al., 2019). Regulations around the world have been limiting soot emissions since the 1970s. The International Civil Aviation Organization (ICAO) until recently limited only the 'smoke number', intended to control visible smoke from aircraft engines which caused dangerous reductions in visibility around airports (George et al., 1972). Modern engines have no visible smoke but still produce invisible nanoparticles (Durdina et al., 2017). In 2020, smoke number was replaced with a limit on the mass concentration of non-volatile Particulate Matter (nvPM) and in 2023 an additional limit was placed on the number concentration of nvPM for all new engines with a rated thrust greater than 26.7 kN (ICAO, 2017). The regulatory term nvPM refers to particles that remain solid when heated to 350 °C. In aircraft emissions, this is primarily soot and concentrations are measured with instruments designed for soot with a low OC/TC ratio (Lobo et al., 2015b) so the terms nvPM and soot will be used interchangeably. Furthermore, regulations on aircraft emissions apply only to turbofan and turbojet engines with rated thrust > 26.7 kN. Volatile particles, lubrication oil particles and secondary organic aerosol may also have important health and climate impacts however, they are not currently regulated and so will not be covered here. Thus, jet engine manufacturers must design new engines to meet the new nvPM standards without exceeding the regulations limiting nitrogen oxides (NOx), unburned hydrocarbons (UHC) or carbon monoxide (CO) emissions while still maintaining strict safety standards. These regulations are aimed at improving local air quality, so engines are assessed based on a standardized landing and take-off (LTO) cycle most relevant for emissions near the ground.

Soot emissions can impact the climate by warming the atmosphere through direct Radiative Forcing (RF) and indirectly by altering cloud processes and decreasing snow albedo (Bond et al., 2013). Aviation is unique in that it emits soot at high altitude with very different atmospheric conditions (e.g., temperature and pressure) from

those on the ground. This may influence the formation of contrails (Kärcher, 2018). Lee et al., (2021) estimated
the climate forcing contribution of carbon dioxide ($CO_2$), contrail cirrus, NOx, soot aerosols, $SO_2$ aerosols and
water vapor from aviation in 2018. By these estimates, contrails account for 57.4 mWm$^{-2}$ or 55% of aviation's net
radiative forcing but with 95% confidence intervals from 27 – 67% of the net RF illustrating the high uncertainty.
The exact RF of contrail cirrus depends on the atmospheric conditions along the flight track and time of day. At
night, contrails have an exclusively warming effect while during the day there can be a warming and a cooling
effect (Stuber et al., 2006).
The estimate of direct RF from soot was relatively low, 0.9 mWm$^{-2}$ (Lee et al., 2021). However,
inventories of global soot emissions from aircraft can vary by two orders of magnitude (Agarwal et al., 2019).
Present inventories are based on the LTO cycle which focuses on landing and take-off at sea-level rather than high-
altitude cruise. As these emissions are measured only at ground level for the LTO cycle, the emissions most relevant
for climate considerations are only indirectly estimated (Stettler et al., 2013). Estimates of emissions inventories
must convert values measured at the ground to account for the drastically different atmospheric conditions at cruise
(Teoh et al., 2024). In addition, the LTO cycle does not exactly match the real time at each thrust for example, the
LTO cycle assumes idle/taxi is 7% but real aircraft use between 3 – 17% thrust for these conditions (Masiol and
Harrison, 2014). Estimates of the RF of soot are from climate models which may underestimate the contribution
of soot (Kelesidis et al., 2022). While $CO_2$ remains in the atmosphere for 100 years or more, soot and contrails
have short atmospheric lifetimes on the order of a week (Bond et al., 2013) or hours (Bock and Burkhardt, 2016),
respectively, so their global warming potential is most important in the short term. This presents an opportunity to
make immediate reductions in global warming and 'buying time' for the implementation of technologies to lower
$CO_2$ emissions (Montzka et al., 2011). This may be important for the aviation industry which in 2022, adopted an
ambitious goal of net-zero carbon emissions by 2050.
These uncertainties highlight the importance of further research to better quantify the role of soot in both
contrail formation (Marcolli et al., 2021) and direct radiative forcing (Kelesidis et al., 2022). In particular, the role
of soot in contrail formation is still unclear and there is high uncertainty in the RF of aviation aerosol-cloud
interactions (i.e. indirect RF) and therefore no best estimate is given by Lee et al. (2021). Such uncertainties make
it difficult to accurately assess priorities in emission reductions as there are often trade-offs between emissions.
For example, there is a well-established trade-off between soot and NOx in diesel engines (Kim et al., 2009) which
has also been observed in aircraft combustors (Harper et al., 2022). Similarly, contrail formation can be avoided
by diverting flights to airspace with unfavorable conditions for contrail formation (e.g. warmer temperatures) but
may result in higher fuel consumption and, thus, $CO_2$ emissions (Teoh et al., 2020). The large uncertainty
associated with the contribution of soot to climate change is in part due to the oversimplification of soot
morphology and composition in climate models which typically assume soot to be coated spheres (Kelesidis et al.,
2022). In reality, soot is an agglomerate composed of polydisperse primary particles (PP), illustrated in Figure 1,
with a nanostructure of layered graphene sheets (Fig. 1: inset).

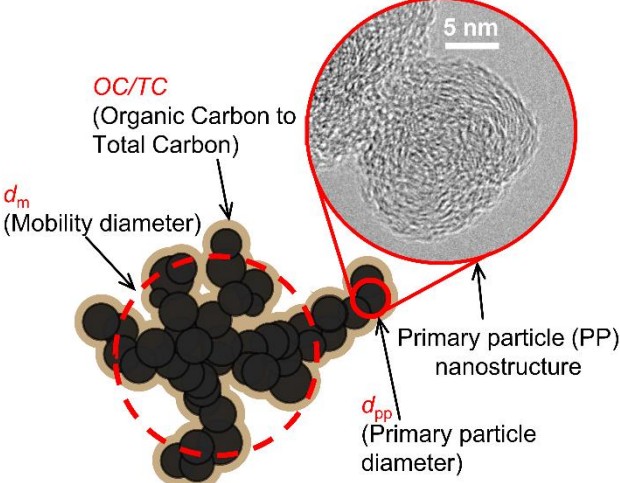

**Figure 1: A schematic of a soot nanoparticle highlighting commonly quantified properties which are relevant for**
**assessing the health and climate impact of such particles including the mobility diameter, $d_m$ (broken line), primary**
**particle diameter, $d_{pp}$ (solid line) and Organic Carbon (brown shaded area) to Total Carbon ratio, OC/TC. The inset**
**shows a high-resolution transmission electron micrograph (HRTEM) of a soot primary particle, from enclosed spray**
**combustion of jet fuel produced at an equivalence ratio of 1.25 (Trivanovic et al., 2022), where the individual graphene**
**layers can be seen. Volatile compounds that may be adsorbed on the surface usually evaporate under the vacuum of the**
**microscope so cannot be visualized easily with HRTEM.**

The relative amounts of Organic Carbon (OC) or Elemental Carbon (EC) compared to the Total Carbon (TC) is typically used to quantify the chemical composition of the particles. The OC is defined by the ICAO as "…carbon volatilized in Helium while heating a quartz fiber filter sample to 870 °C during thermal optical transmittance analysis including char formed during pyrolysis of some materials". Conversely, EC is "…light absorbing carbon that is not removed from a filter sample heated to 870 °C in an inert atmosphere during thermal optical transmittance analysis, excluding char" (ICAO, 2017). The OC/EC ratio is important for source apportionment of ambient aerosols (Ramadan et al., 2000), attempting to understand the health effects of soot (Kelly and Fussell, 2012) and for determining the light absorption of soot (Kelesidis et al., 2021). However, the split between EC and OC is method-dependent (Cavalli et al., 2010) rather than a discrete property. The size of irregular agglomerates such as soot is quantified by equivalent diameters for example, electrical mobility diameter, $d_m$ (Fig. 1: broken line), aerodynamic diameter or projected area equivalent diameter where the type of equivalent diameter depends on the measuring principle. Such agglomerate diameters can be several times larger than the mass-equivalent diameter typically calculated by models (Eggersdorfer and Pratsinis, 2014). Using a realistic soot morphology rather than equivalent spheres in climate models increases the estimated direct RF by 20% on average revealing large direct RF = 3 – 5 W/m$^2$ in hot spot earth regions, in line with field observations (Kelesidis et al., 2022).

Furthermore, limited access to real jet engines has made it difficult to assess the efficiency of soot to act as ice condensation nuclei (ICN) and thus to enhance contrail formation although recently there have been efforts to assess the ICN activity of soot from modern in-use commercial engines (Testa et al., 2024). To date, experiments on the ICN activity of soot have been done primarily using commercial carbon blacks or miniCAST soot generated by burning hydrocarbon gases (Gao et al., 2022). MiniCAST particles tend to have much larger $d_m$ (> 100 nm) than that produced by real aircraft (< 100 nm) if the organic carbon to total carbon ratio (OC/TC) is sufficiently small (Durdina et al., 2016). Recently, enclosed spray combustion of jet A1 fuel has been shown to be a promising laboratory surrogate for aircraft soot produced at high thrust (i.e. cruise) with sufficiently small $d_m$ and OC/TC (Trivanovic et al., 2022). This is important for the calibration of optical instruments which may be sensitive to the OC/TC ratio in addition to particle morphology (Durdina et al., 2016).

Technology for battery-electric or hydrogen-powered planes will not be available in the short-to-medium term for long-haul flights (Schäfer et al., 2019). Significant investment in airport infrastructure would be needed to accommodate such changes in technology (Agnolucci et al., 2013). Emissions from aviation need to be addressed urgently to meet climate goals and prevent further health degradation and mortality from air pollution. However, aircraft engines have many competing demands including continued reduction of gaseous emissions, $CO_2$ net-zero goals, safety requirements and regulations on noise. Thus, a firm understanding of the environmental and health impacts of soot as well as a fundamental understanding of its formation and growth in aircraft engines is essential for weighing the costs and benefits of mitigation strategies. As regulations apply only to turbofan and turbojet engines with rated thrusts > 26.7 kN, most scientific research has been conducted on engines in this category and will also be the category discussed in this paper. However, it is worth noting that small business jets with thrusts < 26.7 kN may produce more nvPM emissions than large aircraft such as the Boeing 737 which do fall under the ICAO regulations and need further research for accurate emissions inventories (Durdina et al., 2019). In addition, leaded aviation gasoline (Avgas) is responsible for lead-containing aerosols internally or externally mixed with soot. While the European Union (EU) voted to ban leaded Avgas used in small piston-engine aircraft in 2022, most other countries still allow its use and it is now considered one of few major sources of ambient lead in the US (National Academies of Sciences Engineering and Medicine, 2021). Possible mechanisms for the formation and dynamics of soot from regulated jet engines will be discussed here. Then, strategies already in use or under development for the elimination of jet engine soot emissions will be reviewed.

## 2. Formation and dynamics of aircraft soot

Although aircraft combustor design can vary significantly, the soot produced by aircrafts have some morphological and compositional differences from other sources such as diesel engines. Aircrafts tend to produce soot with median $d_m$ in the range of 8 (Durdina et al., 2021) to 60 nm (Abegglen et al., 2015). Such small $d_m$ are associated with greater lung deposition efficiency (Rissler et al., 2012) and translocate from the lungs to other organs more effectively than particles with $d_m$ > 100 nm (Cassee et al., 2013). The OC/TC tends to be quite low (< 25%) (Marhaba et al., 2019) when the aircraft operates at high thrust (> 50%) while the reverse is true at low thrust. The OC/TC influences the optical properties of soot and thus its RF (Kelesidis et al., 2021) and will impact the output of optical instruments used to measure aircraft emissions (Durdina et al., 2016). Aircraft soot has PP diameters, $d_{pp}$ (Fig. 1: solid line), from approximately 5 (Liati et al., 2019) up to 24 nm with lower thrusts tending to produce smaller $d_{pp}$ (Liati et al., 2014) which influences soot reactivity (Messerer et al., 2006) and optical properties (Kelesidis et al., 2020). These same properties are also influenced by PP nanostructure which is related to their

maturity (Baldelli et al., 2020). Aircraft tend to produce rather disordered soot with a turbostratic structure with
more defects on its surface than the bulk (Parent et al., 2016). The conditions under which soot forms determine
its final morphology and composition and *vice versa* (Vander Wal et al., 2010).
Figure 2 depicts the cross-section of a single annular aircraft combustor (SAC), one of the common
combustor designs in modern engines. The combustor is typically an annular tube that receives high pressure air
from the compressor, adds energy to the system through combustion and uses it to drive the turbine. Liquid jet fuel
is injected at one end of the SAC, typically with a swirling mechanism to atomize the fuel, promoting evaporation.
However, perfect mixing is not achieved. So locally fuel-rich pockets allow for soot formation even if the global
mixture is fuel-lean. Where the fuel is injected, there is significant recirculation allowing soot to grow in these
fuel-rich pockets (Gkantonas et al., 2020). When there is insufficient oxygen for complete conversion to carbon
dioxide, fuels decompose into radicals and intermediate species, such as acetylene which then grow into small
aromatics (Wang, 2011). These aromatic compounds eventually evolve into polycyclic aromatic hydrocarbons
(PAHs) which are the key gaseous precursors to soot (Frenklach, 2002). The presence of these soot precursors has
been confirmed experimentally with atomic force microscopy (Commodo et al., 2019). With respect to aviation,
experimental studies have shown a correlation between jet fuel aromatic content, sooting tendency (Yang et al.,
2007) and nvPM emissions (Brem et al., 2015). So, fuel composition plays a key role in the formation of soot and
thus provides one possible route for its elimination as discussed in detail in the next section.

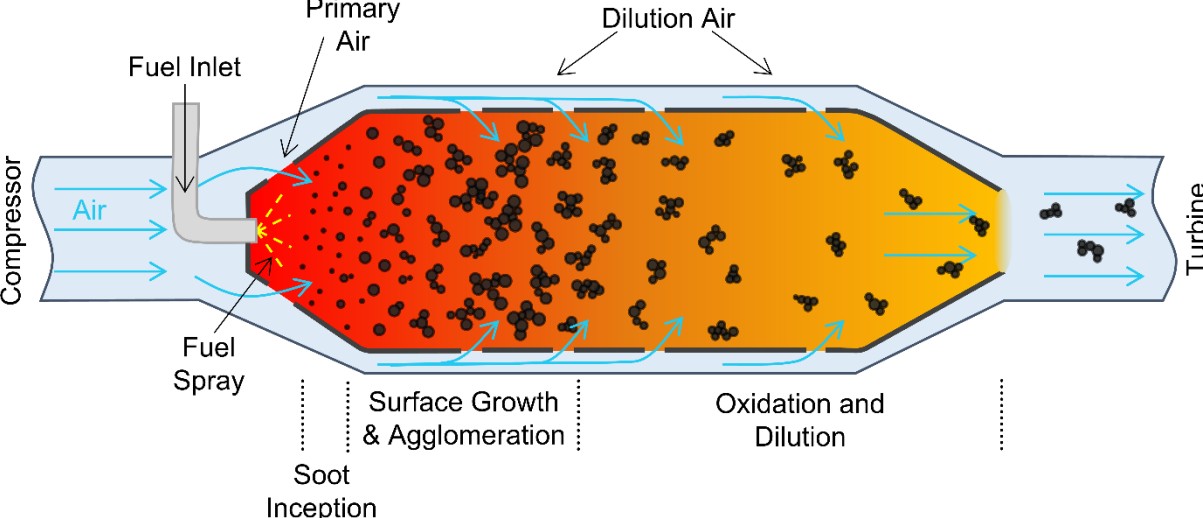

**Figure 2: A simplified schematic of an aircraft single annular combustor (SAC) adapted from (Foust et al., 2012) with a**
**qualitative depiction of the soot dynamics from soot inception to surface growth & agglomeration and then oxidation**
**before being vented to the turbine and eventually the exhaust.**
Although the exact mechanisms of soot nucleation (i.e. the transition from the gas to solid phase) are still an
area of active research (Carbone et al., 2023), the dynamics of soot inception (Sharma et al., 2021) and growth
from nascent to mature soot (Kelesidis et al., 2017b) leading to its final structure are becoming better understood.
Nascent soot particles are as small as $d_m \sim 2$ nm (Camacho et al., 2015), amorphous (Commodo et al., 2017) and
liquid-like (Kholghy et al., 2013) with a carbon to hydrogen (C/H) ratio < 2 (Schulz et al., 2019). As they age,
nascent soot carbonizes (loses hydrogen) and solidifies (Dobbins, 2002). Soot then simultaneously undergoes
surface growth and agglomeration (Kelesidis et al., 2017a). Surface growth of soot is well described by the
hydrogen-abstraction carbon-addition (HACA) mechanism (Frenklach, 2002) although other pathways have also
been proposed (Wang, 2011). During the first few milliseconds of particle growth, surface growth precursors are
depleted then agglomeration takes over as the primary growth mechanism and $d_m$ increases markedly while $d_{pp}$
stays approximately constant (Kelesidis et al., 2017a). In the free molecular regime, particles grow into large
agglomerates through ballistic cluster-cluster coagulation while in the continuum regime this becomes diffusion-
limited cluster agglomeration. Particles which coagulate in the free molecular regime have a slightly more compact
structure than those in the continuum regime as shown by their asymptotic mass fractal dimensions of 1.91 and
1.78, respectively (Goudeli et al., 2015).
This soot growth sequence has been observed and quantified for soot formation in premixed flames, diesel
engines, miniCAST soot generator (Kelesidis et al., 2017b) and even for enclosed spray combustion of Jet A1 fuel
resulting in aircraft-like soot (Trivanovic et al., 2023). After primary air injection for the initial combustion,
dilution air is added at various locations along the combustor length. This oxidizes a sizable portion of the soot
which was initially created. Transmission Electron Microscopy (TEM) has shown that aircraft soot is significantly
oxidized and the small $d_m$ may be in part due to fragmentation of larger agglomerates after extensive oxidation
(Vander Wal et al., 2014). So, in the early stages of the combustor the number and size of soot is likely larger than
what is eventually emitted. The final morphology of the particles, including the $d_{pp}$, $d_m$ and number of PPs per
agglomerate, $n_p$, depends on the initial volume fraction, residence time, temperature and pressure (Kelesidis et al.,
2023a).
While conditions can vary significantly depending on the engine, soot in an aircraft combustor
experiences both high temperature and pressure. In addition, pressures are increased at high thrust which has been
correlated with increased soot concentration and size (Chu et al., 2023). Higher pressures improve the efficiency
of engines and so as engine materials have been improved to withstand higher pressures, the pressure ratios in
engines have also increased. So, soot may begin growing in the free molecular regime but enters the transition
regime as it grows, in particular at high thrust, when pressures are the highest and soot particles tend to grow to
the largest sizes. This is in line with mass-mobility measurements of aircraft soot which shows an increase in the
mass-mobility exponent, $D_{fm}$, from $1.86 \pm 0.37$ to $2.79 \pm 0.07$ as thrust increases from 7 to 118%, respectively
(Abegglen et al., 2015). However, mass-mobility measurements are not part of the regulatory framework for
aircraft nvPM.
Low thrusts lead to the longest soot residence time in the combustor but tend to produce the smallest
particles both in terms of $d_m$ and $d_{pp}$ which can be attributed to the smaller amount of fuel resulting in a lower
volume fraction of nascent soot (i.e. less nucleation) and allowing for a longer residence time in oxygen rich zones
which oxidizes the soot reducing both the number and size of particles (Durdina et al., 2014). At the same time,
the OC/TC increases at low thrust which could be attributed to poorer combustion efficiency at these conditions.
At high thrust the residence time is short but initial number concentrations are higher due to high fuel flow. The
time in oxidating zones is reduced also, resulting in a larger number concentration, $d_{pp}$ (Liati et al., 2014) and $d_m$
(Abegglen et al., 2015). Simulations of aircraft combustors have shown that soot forms intermittently in locally
rich regions of the flame and, due to recirculation, soot spends $4 - 5$ times longer in the combustor than the fluid
time scales (Chong et al., 2018b). The high-temperature residence time of soot in a combustor can only be
estimated from simulations that account for the geometry, fluid flow rates, temperature and pressure in a given
combustor.
Modeling soot emissions accurately remains a challenge (Chong et al., 2018a) because soot formation in
combustors is intermittent. So, simulations must take place over a large time frame to achieve a statistically
representative time-averaged result (Franzelli et al., 2023). Furthermore, the transport and chemistry of soot must
be solved simultaneously in order to capture the real volume fraction, $f_v$, and particle size distributions (PSD)
(Gkantonas et al., 2020). The most detailed simulations to date have utilized laboratory combustors such as the
Cambridge Rich Quench Lean (RQL) burner (Gkantonas et al., 2020). These laboratory burners are optically
accessible for laser diagnostics allowing for a detailed comparison to the evolution of soot $f_v$ and PSD. However,
the laboratory burners use ethylene, a gas, instead of liquid jet fuel and pressures are up to 5 bar (Chong et al.,
2018a). Modern aircraft engines may have pressures up to an order of magnitude higher than this at certain
conditions (Nguyen et al., 2019). Nonetheless, such simulations can give insight into the formation and growth of
soot in aircraft combustors capturing some of the trends observed experimentally. Specifically, simulations show
that soot forms near the shear layers between the fuel and oxidizer streams and then enters an inner recirculation
zone where it grows further (Gkantonas et al., 2020). Fuel rich pockets can also break off from the main jet and
become entrained in the recirculation zone driving the intermittent soot growth within the combustor (Chong et
al., 2018a). Soot was shown to grow by both acetylene-based surface growth (e.g., HACA) and condensation via
aromatics (Gkantonas et al., 2020). Simultaneously, significant oxidation reduces the particle size and can induce
fragmentation increasing the number concentration (Gkantonas et al., 2020) which is supported by experimental
data (Vander Wal et al., 2014). Introduction of dilution air part way through the burner oxidizes soot in the lean
combustion zone as well as lowers the rate of soot formation near the nozzle (Chong et al., 2018a). Higher pressures
in the model combustor result in larger soot $f_v$, a trend which was captured by simulations but the total $f_v$ for the
high pressure condition was underpredicted by a factor of 4 (Chong et al., 2018a). Therefore, simulations can give
insight into the formation of soot in aircraft combustors but significant improvements are needed to have truly
predictive models which can aid in combustor design (Franzelli et al., 2023). It is worth noting that these
simulations focus on capturing the number and mass emissions from combustors, but do not seem to account for
the realistic morphology of soot particles which are highly irregular agglomerates rather than spheres. The
assumption that soot is spherical rather than an agglomerate with polydisperse primary particles can significantly
change the resulting estimate of soot $d_m$, number and, most importantly, $f_v$ (Kelesidis and Goudeli, 2021).
Therefore, it is not realistic to predict soot formation without properly accounting for its shape (Bouaniche et al.,
2020). For example, models overpredict soot volume fraction by up to 3 times when particles are assumed to be
spherical (Kelesidis and Pratsinis, 2021). Such realistic descriptions have, for example, been used quite effectively
to describe black carbon (BC) formation and growth from a variety of combustion sources and even facilitate
monitoring of BC emissions by aerosol (e.g., particle mobility and mass analyzers), laser (e.g., light extinction)
diagnostics and fire detectors by accounting for BC morphology and limiting the current uncertainty regarding BC
mass and particle size (Kelesidis et al., 2020). In addition, by capitalizing on the accurate description of the high
temperature residence time during enclosed combustion synthesis of nanomaterials and the latest advances in soot
structure and composition, more than 99% of the emitted soot mass and concentration from enclosed jet fuel
combustion was removed (Kelesidis et al., 2023b).

## 3. Means for the elimination of aircraft soot

### 3.1 Sustainable aviation fuels

Alternative aviation fuels include any fuels aside from kerosene-based jet fuels and Avgas. This includes, for
example, hydrogen, ammonia and jet fuels made without fossil fuels. Sustainable Aviation fuels (SAF) are non-
fossil jet fuels that are attractive due to their potential to act as a drop-in solution for reducing $CO_2$ emissions as
these fuels can be used directly in existing engines. The ICAO specifies fuels must be "completely interchangeable
and compatible with conventional jet fuel" in order to prevent the safety risks of mishandling and high costs of
additional infrastructure (ICAO, 2018). Alternative jet fuels can be considered 'drop-in' when they do not require
new fuel systems, distribution networks or new aircraft (ICAO, 2018). Sustainable Aviation Fuels (SAF) are
mainly produced from biological feedstocks (e.g. soybeans, sugarcane, biomass, etc.) (Staples et al., 2018). These
are converted into liquid hydrocarbon fuels through processes such as Hydroprocessed Esters and Fatty Acids
(HEFA), Fischer-Tropsch (F-T) or Alcohol-to-Jet (ATJ) to name a few (Brooks et al., 2016). Similarly, e-fuels use
$CO_2$ capture and sustainable energy sources such as solar to produce synthetic jet fuels (Schäppi et al., 2022).
Currently, SAF are only certified for use when blended with conventional jet fuel although efforts are being made
to certify 100% SAF in the future. Flights powered with 100% SAF have already been performed for research
purposes (Märkl et al., 2024). The $CO_2$ reduction from such fuels comes primarily from the synthetic or biological
$CO_2$ captured during the production process. Actual $CO_2$ released from the engine remains about the same as
conventional jet fuel. So, a Life Cycle Analysis (LCA) is needed to account for the so-called Well-to-Wake
emissions (Han et al., 2013). The total reduction in Green House Gas (GHG) emissions will depend on both the
GHG emissions associated with production of the petroleum based jet fuel as well as the net GHG emissions from
growing, transporting and burning the SAF fuels. The ICAO, under the Carbon Offsetting and Reduction Scheme
for International Aviation (CORSIA), certifies alternative fuels as SAF based on a standardized LCA. While the
exact reduction in GHGs will change as technologies evolve, an LCA of the best case scenarios show up to a 68%
reduction in $CO_2$ emissions if SAFs account for > 85% of all aviation fuels (Staples et al., 2018).
In addition to reducing net-$CO_2$ emissions, SAFs also have the potential to reduce soot emissions and thus
the health impact and non-$CO_2$ radiative forcing of aircraft emissions which is typically excluded from LCA
analysis (Staples et al., 2018). These fuels tend to have a lower aromatic content than fossil fuels which has been
correlated to the number of particles emitted by an aircraft (Brem et al., 2015). As discussed previously, aromatic
species are key precursors to soot formation and thus a decrease in fuel aromatics may reduce the rate of soot
nucleation. The hydrogen-to-carbon ratio (H/C) of the fuel has been shown to have an even greater anti-correlation
with aircraft soot emissions than fuel aromatic content (Brem et al., 2015). While H/C has long been associated
with the sooting tendency of a fuel (Yang et al., 2007), the mechanism for this is less clear as it is difficult to
separate from effects such as lower flame temperatures (Xue et al., 2019). Blends of a HEFA-based SAF with Jet
A1 up to 50% (the current upper limit for a SAF blend) showed a ~35% reduction in number based nvPM and
~60% reduction in mass based nvPM (Lobo et al., 2015a). These reductions correlated best with the H/C content
of the blends. In fact, the geometric mean $d_m$ has been shown to drop nearly linearly as H/C increases while
decreases in nvPM number were not as steep suggesting that the decrease observed in nvPM mass is strongly
influenced by the smaller particle sizes for HEFA-based fuels (Durand et al., 2021). Similar trends were observed
for a range of different SAF types including HEFA, Alcohol-to-Jet (ATJ) and a Catalytic Hydrothermal Conversion
Jet (CHCJ) fuel showing that the dependence on H/C is not dependent on the fuel production method (Harper et
al., 2022). The size distributions of the soot produced shifted to smaller mobility diameters from $d_m$ = 49 to 22.5
nm and narrowed the distribution from a geometric standard deviation, $\sigma_g$ = 1.99 to 1.58 with pure Jet A1 and a
50% blend, respectively (Lobo et al., 2015a). With pure Jet A1, the $\sigma_g$ approaches that of the self-preserving limit
for agglomerates coagulating in the free-molecular regime (Goudeli et al., 2015) while the $\sigma_g$ produced with the
SAF blends are significantly smaller. This could be due to the decreased number concentration from extended
surface growth and less agglomeration. As the aromatic content and H/C of conventional jet fuels varies, the actual
reductions achieved with SAF blends will depend on composition of the conventional jet fuel in the blend,
particularly when the SAF blending ratio is low. Currently, alternative fuels are designed primarily with the goals
of reducing life-cycle $CO_2$ emissions and matching the properties of conventional jet fuels. However, there is an

opportunity to also optimize jet fuel composition for minimum soot emissions. Schripp et al., (2021) showed that different SAF could be blended to obtain a desired H/C, while maintaining regulatory specifications for jet fuels. Soot emissions of these fuels were first tested in a laboratory flame, then the optimal mixture was used in a real jet engine to confirm the trends seen in the laboratory resulting in emission reductions of particle mass and number by 29 and 37%, respectively, when using a 38% SAF blend with Jet A1 (Schripp et al., 2021). However, if a SAF blend is designed with higher aromatic content and lower H/C than a conventional fuel, soot emissions could even increase (Schripp et al., 2019). Laboratory tests are essential for speeding up the design of alternative fuels since real jet engines are inaccessible to many researchers and too costly to operate for initial screening tests. A standardized flame for assessing the sooting properties of jet fuels would assist in the development of alternative fuels however, there is currently no standardized method for such experiments. Enclosed spray combustion is a promising unit for such in lab approaches (Trivanovic et al., 2022).

Several publications have shown that the benefits of a SAF blend are thrust-dependent. For example, a 32% blend of HEFA-synthetic paraffinic kerosene and Jet A1 at idle operation showed a 60 and 70% reduction in number- and mass-based nvPM, respectively (Durdina et al., 2021). The same blend at 65% thrust resulted in only a 12% reduction in number-based nvPM and at take-off the reduction was only 7%. In this case, the use of such SAF blends may improve local air quality by reducing emissions in the vicinity of airports but may not make a significant impact on cruise conditions which are most concerning for climate change. It is worth noting that the majority of studies on aircraft soot emissions are done at ground level which has significantly different atmospheric conditions than cruise in the upper atmosphere. Ideally, cruise emissions should be measured behind an aircraft in-flight, but this is rarely done due to the cost and logistical challenges. One of the few in-flight studies comparing conventional jet fuel to a 50% HEFA blend showed a 50 and 70% reduction in particle number and mass emissions, respectively, behind an aircraft with a medium thrust setting of ~ 50% (Moore et al., 2017). At the high thrust setting, the particle number reduction was only 25% (Moore et al., 2017), supporting the trend observed on the ground. The wide range of values listed here highlights the need for more studies both at the ground level and in-flight.

Currently, SAF must be blended with conventional jet fuel (up to 50%) for safety reasons although 100% blends may be allowed by 2030. In practice, supply issues keep the use of SAF low accounting for an estimated 0.1 – 0.15% of global jet fuel use in 2022 despite a tripling in the supply of SAF from 2021 to 2022. If the SAF supply is limited and individual flights only have a very small fraction of SAF in the fuel, there will likely be no effect on the soot emissions (Lobo et al., 2015a). So, while alternative fuels could provide a short-term solution to reducing aircraft emissions, the speed at which this is adopted is still limited. Targeted use of the limited SAF supply could be used in the short term to maximize the benefits of such fuels while supply is limited. For example, contrails with the greatest warming effect are commonly at dusk during the winter (Teoh et al., 2022a) so fueling flights at such times with high SAF blends could have the biggest benefit. One analysis found that compared to a 1% SAF blend for all transatlantic flights, fueling the 2% of flights producing the highest RF with a 50% SAF blend could take the total RF reduction from 0.6% up to 6% (Teoh et al., 2022b). The European Commission and the US have implemented policies to mandate the annual uptake of SAF which may prohibit the targeted use of SAF. For example, starting in 2025 it will be required that "all aviation fuel supplied to aircraft operators at (European) Union airports contains a minimum share of SAF" (European Commission, 2021). Further press releases confirm that "this means that every flight leaving the larger EU airports, will carry a minimum amount of SAF" (Fit for 55 and ReFuelEU Aviation). Some airlines have made similar pledges for example, Air France KLM promises they "will add a percentage (0.5% to 1%) of SAF on all flights departing from France and the Netherlands" (Air France KLM Sustainable Aviation Fuel). Thus, while the supply of SAF is limited, it will be used in more aircraft at lower blending ratios missing an opportunity to reduce soot emissions. Intelligent changes to policy on the use of alternative fuels could thus reduce the net-RF of aviation without needing to increase the supply of SAF.

3.2 Aircraft Combustor Design & Operation

The limitations of alternative jet fuels highlight the continued need for improved and novel engine technologies which could be used also with alternative fuels to minimize the total impact of aviation on the environment. Here, only combustion engines will be considered as electric aircrafts are estimated to account for only a quarter of all passenger-miles in 2050 (Prabhakar et al., 2022). Furthermore, some in-development technologies, such as open rotor engines, promise significant reductions in fuel consumption (Khalid et al., 2013) and soot emissions but are not discussed here as they are not linked to the actual formation of soot. Since nvPM regulations only recently came into effect, most aircraft combustors are designed primarily to lower NOx, but some designs can also reduce soot. Alternative fuels have not been shown to reduce NOx emissions compared to conventional jet fuel (Moore et al., 2017). Combustor designs must balance limits for all regulated gas and particulate emissions, fuel efficiency,

safety and cost. Rich Quench Lean (RQL) combustors have been used by the aviation industry since at least the 1980s to reduce NOx emissions while maintaining sufficient combustion stability (Novic et al., 1983). Today, they are the most common type of combustor listed in the ICAO emissions database (ICAO Aircraft Engine Emissions Databank, 2023). Briefly, RQL combustors have three zones, depicted in Figure 3. First, there is a fuel-rich zone that allows for more stable combustion which is important for the safety of the aircraft. Rich conditions have lower combustion efficiency and promote the formation of soot, UHCs, and CO. In the quenching zone, a large volume of cool air is injected to provide oxygen for completing the conversion of UHCs and CO to $CO_2$ while lowering the temperature to minimize NOx formation. The air flow for the rich combustion stage and quenching zone are controlled separately and further dilution air may be added before the gases are sent to the turbine. Although the mixing and residence times in RQL combustors were originally optimized for reducing NOx (Rizk and Mongia, 1990) proper design and operation can also reduce soot emissions through oxidation during the lean burn stage. In fact, it was shown in a laboratory setting that a judicious injection of fresh oxygen in a manner similar to RQL combustors can promote oxidation of soot removing up to 99.6% of the initial soot volume fraction from jet fuel combustion (Kelesidis et al., 2023b). Such results can be scale up by matching the high temperature residence time as has been shown with scale-up of flame synthesis of nanoparticles from mg to kg per hour (Kelesidis and Pratsinis, 2021). When quenching air is introduced farther downstream in the combustor, soot has more time to form and grow. Hence, oxidation is less effective. Earlier injection of air with sufficient turbulent mixing has the opposite effect, minimizing soot emissions (El Helou et al., 2021). However, if quenching air is injected too early this could increase NOx emissions or reduce combustion stability.

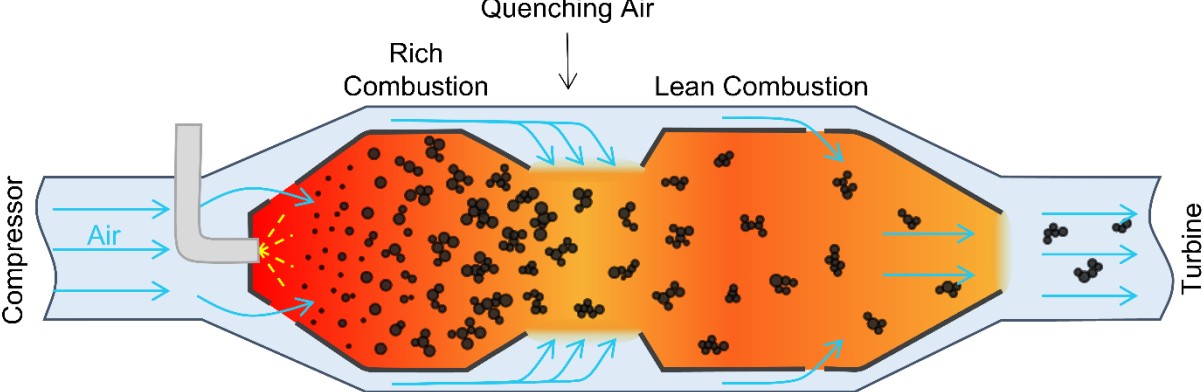

**Figure 3: A simplified schematic of a Rich Lean Quench (RQL) aircraft combustor adapted from (Rizk and Mongia, 1990) where there is first a fuel rich combustion zone, followed by a large flow of quenching air to lower the temperature and dilute to a globally lean combustion zone. The dynamics of soot are qualitatively depicted from inception to surface growth, agglomeration and oxidation.**

In 1995, the first Double Annular Combustor (DAC) was used commercially. This combustor design has two stages as the name implies, depicted in Figure 4. At low thrust (e.g., idle) only the pilot stage is used with a low air to fuel ratio and low flowrate to ensure good ignition and to reduce CO and UHC emissions. When sufficiently high thrust is achieved, both the pilot and main stage are ignited with a high air to fuel ratio (lean burn) and high flowrates (Boies et al., 2015). This had the desired effect of reducing the NOx emissions over the LTO cycle by ~30% compared to a single annular combustor on the same engine (Mongia, 2007). Soot emissions from a DAC equipped engine vary significantly with thrust. At low thrust, when only the pilot stage is ignited, soot emissions are high, and increase with increasing thrust in both number and mobility diameter (Boies et al., 2015). When both stages are ignited at thrust ~25%, the soot concentration and size drops significantly (Boies et al., 2015). Similarly, a DAC using only the pilot stage showed an increased mass concentration of organic particulate matter compared to when both stages were used (Lobo et al., 2015b). The morphology of soot produced in both stages is in the range observed in other combustors.

As demonstrated by the low emissions of DAC when operated in the lean combustion mode, lean burn engines have the potential for extremely low emissions if the combustion stability issues can be overcome. In fact, lean combustion technologies typically produce an order of magnitude less soot than an RQL combustor (Liu et al., 2017). Lean burn combustors were first developed for stationary gas turbines used for energy generation where safety requirements are less strict and are now being transferred to aviation as technology improves. Such technologies include Lean Direct Injection (LDI) or the Multipoint Lean Direct Injection concept (MLDI) (Liu et al., 2017). Direct injection is used to reduce the risk of autoignition that comes with premixed combustion. The use of multiple injectors, depicted in Figure 5, along with intense mixing creates conditions similar to lean, premixed combustion. In an LDI combustor a central pilot injector is surrounded by multiple main fuel injectors

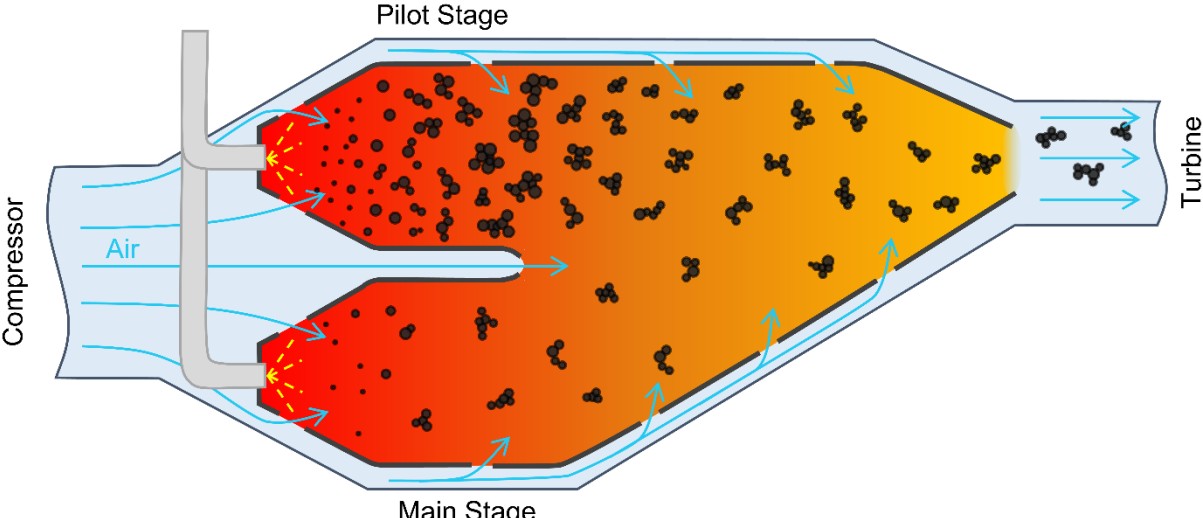

**Figure 4: Simplified schematic of a Double Annular Combustor (DAC) adapted from (Foust et al., 2012) and a qualitative depiction of the dynamics of soot surface growth, agglomeration and oxidation within the combustor.**

with little to no dilution added after the initial air supply near the fuel injectors. The MLDI concept is similar to the LDI combustor with an altered injector layout. Globally lean combustion with good mixing is unfavorable for soot production as there are few locally fuel-rich areas. At the same time, low temperatures from the lean burn reduce NOx emissions significantly (Liu et al., 2017). Regulatory measurements of nvPM emissions from an LDI combustor show nvPM mass and number emission levels on par with RQL combustors with similar rated thrusts (ICAO Aircraft Engine Emissions Databank, 2023). To the best of our knowledge, no studies have characterized the size, morphology or chemical composition of soot from an LDI equipped engine. The limited data for such combustors makes the real emissions performance of such an engine difficult to assess.

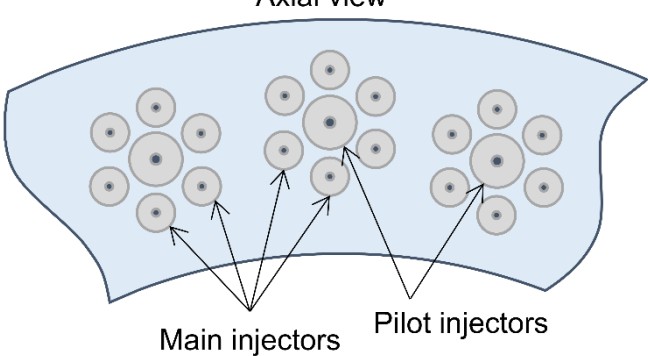

**Figure 5: A simplified schematic of a Lean Direct Injection (LDI) combustor adapted from (Fric, 1995) which features a central pilot injector surrounded by multiple main injectors. These combustors usually have most or all the air flow into the combustor around the fuel injectors without subsequent dilution to provide intense mixing for lean combustion with close to premixed combustion.**

Lean Premixed Prevaporized (LPP) combustors aim to completely vaporize the jet fuel prior to ignition in order to have lean, premixed combustion (Figure 6). Without locally fuel-rich conditions, little to no soot will form. As with the LDI combustors, there is little dilution after the initial injection of primary air for combustion. Premixed combustion with high pressures comes with a risk of autoignition in the mixing zone so careful design of the combustor is needed to prevent such instabilities. These combustors use special fuel injectors to achieve near-premixed lean combustion conditions which tend to form significantly less soot. Both the LDI and LPP combustor designs achieve stable combustion through complex combustor design which could lead to increased cost and maintenance. So, lean conditions are favorable for emissions reduction but come with engineering challenges. Theoretically, new jet fuels with lower lean blow-off (LBO) limits could extend the lean operating range of an engine and conversely, fuels with an insufficient LBO could pose a safety risk (Undavalli et al., 2023).

Recently, a novel research engine called the Lean Azimuthal Flame (LEAF) combustor (not yet in commercial use) using "flameless oxidation" has been developed for soot-free and low NOx combustion (Oliveira et al., 2021). This concept can be further improved through co-combustion of small amounts of hydrogen which extends the operating window (Miniero et al., 2023). The use of hydrogen helps to stabilize the combustion without the use of a fuel-rich pilot flame that can increase soot production as with the DAC combustors. Such concepts

which require an additional fuel that cannot be used in all engines require significantly more capital to implement
because additional infrastructure needs to be built to support, for example, hydrogen storage and fueling.
Furthermore, such parallel infrastructure poses a safety risk if an aircraft is filled with the wrong fuel and therefore
such solutions are not promoted by the (ICAO, 2018). So, combustors which achieve lean, premixed conditions
are promising for achieving both low soot and low NOx emissions but pose design challenges.

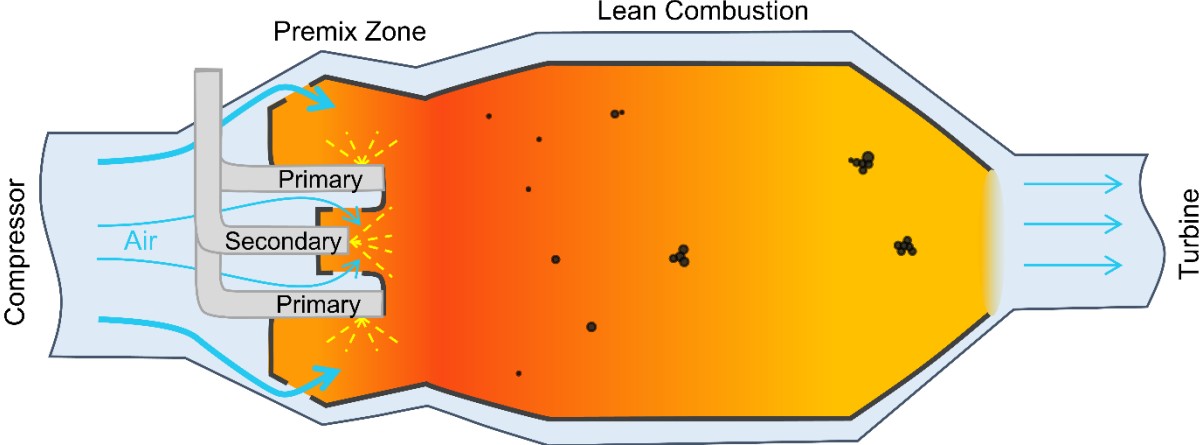

**Figure 6: A simplified schematic of a Lean Premixed Prevaporized (LPP) combustor adapted from (Foust et al., 2012)**
**which contains multiple injectors that spray fuel into the premix zone where the jet fuel completely vaporizes without**
**ignition. Then, in the combustion zone the premixed fuel is ignited under fuel-lean conditions which nearly eliminate**
**soot while low temperatures prevent the formation of NOx.**
The ICAO provides a public database of regulated emissions with the earliest nvPM emission test dates
starting in 2014 (ICAO Aircraft Engine Emissions Databank, 2023). These data are collected and reported by the
engine manufacturers following the standards laid out in the ICAO Annex 16 for engine emissions certification
(ICAO, 2017). Emissions are tested across the entire LTO cycle which includes idle/taxi (7% thrust), approach
(30%), climb-out (80%) and take-off (100%) for both nvPM mass and number. Data submitted to the ICAO
database should be collected following the procedure outlined in the ICAO Annex 16, Vol. II (ICAO, 2017).
Briefly, particles are sampled at the engine exhaust with a no more than 35 m long (from probe tip to instrument
inlet) heated sampling line to the measurement devices. This relatively long line, paired with the small size of
aircraft soot may result in significant diffusional and thermophoretic losses due to temperature gradients as the
sample cools from the exhaust temperature to sample line temperature. Since 2017, the nvPM mass and number
diffusion and thermophoretic losses must be accounted for with the methods outlined in the ICAO Annex 16, Vol.
II (ICAO, 2017). However, it is important to note that these losses are size-dependent, but the regulations do not
require particle size measurements. Therefore, the estimate of the line loss correction may not be accurate for all
engines. Figure 7 shows the nvPM number emissions normalized by the fuel flow (#/kg) at (a) idle/taxi and (b)
take-off for simplicity, although approach and climb-out data are also available (ICAO Aircraft Engine Emissions
Databank, 2023). Mass nvPM data shows similar trends. The main differences between nvPM mass and nvPM
number are the LPP values falling closer to the other combustors with mass-based emissions compared to number
based emissions. This suggests that the LPP produces fewer but larger particles than other combustors on average.
Values for approach and climb-out tend to fall between those measured at the extremes for both number and mass
nvPM. Combustor names are provided for all entries in the database and can be grouped by type if sufficient
information is given by the manufacturer. The RQL combustors make up the majority of reported data (134 entries),
followed by SAC (38), LPP (26), LDI (7) and DAC (2). Within these broad categories, there are multiple distinct
implementations of these combustor types. For example, the RQL category contains the Rolls-Royce Phase5 series,
Pratt & Whitney Talon series and General Electric LEC series combustors. The SAC (squares) have some of the
highest emissions in the database, but a group of SAC are approximately an order of magnitude lower at idle/taxi
(Fig. 7a). These lower emission SAC are modified for better performance (CFM Tech Insertion) which seems to
improve emissions at low (7 and 30%) thrust with little change at high (80 and 100%) thrust. The data for RQL
combustors (circles) have the most variation quite likely due to the variety of different implementations of the
RQL concept compared to all other combustor types. This highlights the fact that RQL burners can have quite low
particulate emissions if designed and operated properly, particularly for engines with lower static rated thrust. At
take-off (Fig. 7b), the LPP combustors (triangles) clearly outperform all other combustors in the database. All of
the LPP combustors are from the TAPS combustor series. At idle/taxi (Fig. 7a) LPP combustors still perform well
but some RQL and modified-SAC combustors have similar or lower emissions. In an LPP combustor, all injectors
are on during high thrust operation and premixed combustion can be achieved resulting in lower emissions.
Conversely, at low thrust only some of the injectors are used to lower the power output without creating conditions
which are too lean for stable combustion which may explain the higher emissions at idle/taxi compared to take-
off. A similar phenomenon has been observed in scientific studies of DAC engines where emissions were reduced
significantly when both combustor stages were in use at approximately thrusts > 30% (Boies et al., 2015). The
small number of entries for DAC and LDI combustors makes it difficult to draw conclusions about such
combustors but the data that are provided for both fall in approximately the middle of the nvPM emission range.
So, at present LPP combustors seem to perform at least as well as other combustors at idle/taxi and significantly
reduce emissions at take-off resulting in the lowest overall emissions in the ICAO database. It is worth noting that
engine operation can also reduce emissions, for example reduced thrust take-off has been shown to reduce fuel
consumption, NOx and black carbon (soot) emissions by 1.0–23.2%, 10.7–47.7%, and 49.0–71.7% respectively
(Koudis et al., 2017).

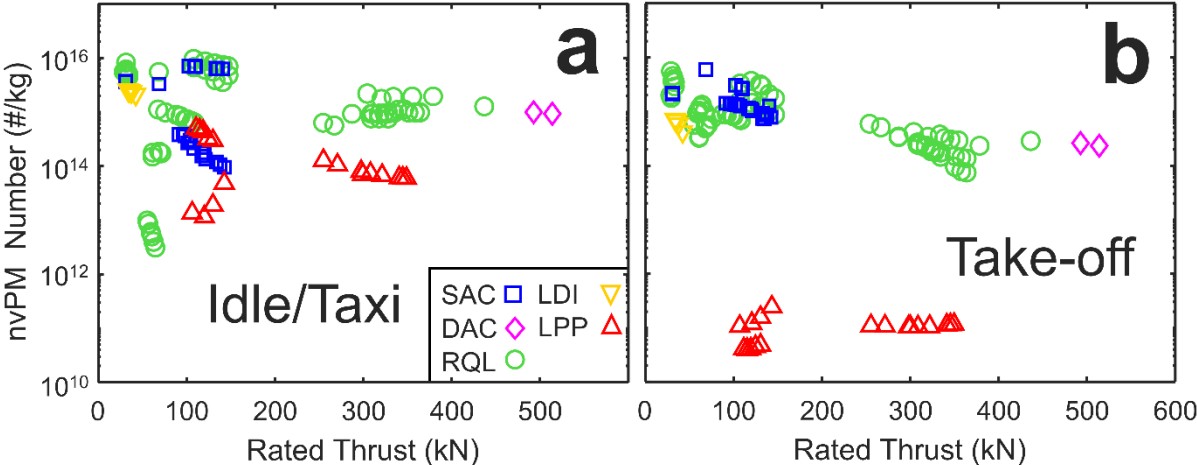

**Figure 7: The nvPM number as a function of an engine's rated thrust at (a) idle/taxi (7% thrust), (b) take-off (100%).**
**Combustor types represented in the database include SAC (squares), DAC (diamonds), RQL (circles), LDI (inverted**
**triangles), LPP (triangles). The total nvPM number is normalized by the fuel flow (kg).**
While the ICAO database provides information on the mass and number of nvPM emissions, it does not
include any morphological or chemical characterization of the particles. Furthermore, the data are collected by the
engine manufacturers, rather than independent researchers although for a small number of engines, research
measurements are also included (ICAO Aircraft Engine Emissions Databank, 2023). Thus far, the vast majority of
academic studies on soot emissions from aircraft engines have been conducted on large commercial aircraft (rated
thrust >26.7 kN) most with SAC combustors (Abegglen et al., 2015; Beyersdorf et al., 2014; Elser et al., 2019;
Johnson et al., 2015; Liati et al., 2014; Marhaba et al., 2019; Parent et al., 2016). A few studies have explored soot
from DAC (Boies et al., 2015; Johnson et al., 2015; Lobo et al., 2015b) and RQL (Brem et al., 2015; Delhaye et
al., 2017; Saffaripour et al., 2017) engines. There is relatively little scientific research on soot emissions from
novel aircraft engines. For example, the Water-Enhanced Turbofan (WET) concept expects to significantly reduce
soot emissions during the stage where water is captured for recirculation (Kaiser et al., 2022). However, to the best
of our knowledge, there are no scientific studies published on the actual measured or modeled soot emissions from
WET engines. The limited number of studies characterizing soot emissions from 'low emission' engine technology
highlights the need for more research on such engines if they will be adopted in the future. Commercial deployment
of new engine technologies takes a significant amount of time and money and so, when a new technology is
deployed it remains in use for many years with the life span of an average aircraft spanning from 20 – 30 years
(Ceruti et al., 2019). This makes it essential to identify which technologies offer the best emissions profile before
it is commercially scaled up, for example through the use of computational fluid dynamics (CFD).
**4. Conclusions**
Soot from aviation has a negative effect on human health and can contribute to climate change through direct
radiative forcing and increasing the formation of persistent contrails. New regulations have been put into place to
limit soot emissions in addition to other pollutants such as NOx, UHC and CO. The strategies for reducing one
type of pollutant may increase another with soot and NOx emissions often at odds with one another. Non-$CO_2$
aircraft emissions are estimated to be two thirds of aviation's net-RF, but the uncertainties associated with the non-
$CO_2$ terms are very high. The difficulty in reducing soot emissions from aviation comes primarily from the
competing requirements which include safety, reduction of gaseous pollutants and cost. A better understanding of
the role of soot and other non-$CO_2$ emissions is needed to properly assess trade-offs between design requirements

and avoid improving emissions of one pollutant while increasing another's or compromising safety. Thus, without a robust understanding of the role of soot in direct and indirect radiative forcing (e.g. through contrail formation) trade-offs between soot reduction and other pollutants cannot be properly accounted for.

Aircrafts tend to produce soot with relatively small $d_m$ which has greater health impacts than larger soot particles. Soot nucleates in locally fuel-rich zones (created by the jet fuel spray) then grows through surface growth, condensation and agglomeration (Trivanovic et al., 2023). The OC/TC ratio of aircraft soot, which has implications for the source apportionment (Ramadan et al., 2000), health effects (Kelly and Fussell, 2012) and optical properties of soot (Kelesidis et al., 2021), depends on thrust (Elser et al., 2019; Fig. 6). Low thrust is associated with high OC/TC and high with low OC/TC. Extensive oxidation reduces the number concentration and size of soot resulting in smaller particles than other combustion sources (e.g. diesel). Significant progress is still needed to accurately quantify this process in realistic aircraft combustors. Some progress has been made in recent years matching experimental data from laboratory combustors but there are important differences between laboratory combustors and real aircraft combustors and simulations are not yet able to match the output of these simplified combustors at all conditions. A realistic description of BC allowed for the first time to determine conditions for synthesis of carbon black (CB) with closely controlled structure and size that is crucial for its diverse applications where for tire reinforcement hard agglomerates consisting of large primary particles (PP) are needed as fillers while for battery electrodes such agglomerates should consist of much finer PP and for inks or paints the CB agglomerates should be soft ones (Kelesidis et al., 2023a). Clearly such an understanding should be incorporated into the design of aircraft engines burning fossil and/or sustainable aviation fuels as it greatly facilitates engine design and operation for complete oxidation of any soot formed before its emission. The high cost and 20 – 30 year lifespan of aircraft necessitates robust models to aid in combustor design and operation for further technological advancements.

Sustainable Aviation Fuels (SAF) have the potential to significantly reduce soot emissions due to the typically lower aromatic content and increased H/C ratio typically associated with these fuels in addition to reductions in lifetime $CO_2$ emissions. Although most literature on the use of such fuels does show that it reduces soot emissions, the reduction appears to be thrust dependent. So, it has the greatest effect on reducing low-thrust emissions which are important for local air quality (e.g., idle) although modest reductions have also been observed at high altitude cruising conditions. Several SAFs are approved for commercial use but lack of sufficient supply makes it a tiny proportion of the global jet fuel supply (0.1 – 0.15% in 2022). If SAFs are blended at small proportions with conventional jet fuel, the soot reduction benefits might be hardly seen. Targeted use of high SAF blends on certain flights rather than low SAF blends for all flights could be the best use of a limited resource. Supply issues likely will not be overcome soon, so policies mandating the use of SAF fuels should be designed in a way that encourages the use of a targeted approach that will also lower soot emissions, not just life cycle $CO_2$.

Soot is primarily produced during fuel-rich combustion. So, throughout the years efforts have been made to move toward fuel-lean combustion processes. The RQL combustors use a lean quenching stage after an initial rich burn to ensure good combustion stability while still reducing NOx and in some cases soot. The design of the quenching stage is essential for balancing combustion efficiency, NOx, and soot emissions from such engines. The DAC combustors similarly take advantage of a pilot stage with low air to fuel ratios for combustion stability at low thrust and a second main stage combustor which can be used at medium to high thrust for lean combustion with a high air to fuel ratio. When both stages are in use, DAC combustors have very low soot emissions but when only the pilot stage is used, soot emissions can be higher than in a traditional burner particularly at medium-low thrust (e.g., ~20%). More recently, advances have been made on truly lean engine technologies. This can be achieved either by using multiple injectors and high mixing rates to achieve nearly premixed combustion or through mixing zones which allow for full evaporation of fuel before ignition. These lean burn engines promise the lowest emissions of soot and NOx due to the lower temperatures and lack of fuel-rich zones. High complexity in such burners may result in higher maintenance costs. Finally, hydrogen can be used to help stabilize lean combustion such as in the LEAF combustor which is both soot-free and low NOx but is still under development in academic laboratories. However, the ICAO is discouraging such solutions which require fuels that are not "drop-in" (e.g. hydrogen), as incompatibilities between engines and fuels could pose safety risks and require significant capital investment in infrastructure.

The combined use of fuels with low sooting propensity and operating at lean combustion conditions have the potential to reduce or even eliminate soot emissions from aircraft engines. However, caution should be used whenever there is a trade-off with other emissions (i.e. NOx) as there is still significant uncertainty in the contribution of soot to direct RF and its role in contrail formation. The development of computational models which can accurately predict soot production from various combustor designs and modes of operation will be essential for minimizing soot emissions from aircraft while balancing other considerations. This will rely on further

fundamental research to better understand soot nucleation rates to close the soot mass balance and match field
data.

**Competing interests**

The contact author has declared that none of the authors has any competing interests.

**Acknowledgements**

We gratefully acknowledge Christian Kubsch for proofreading this article. This research was funded by the Particle
Technology Laboratory, ETH Zurich, and in part by Swiss National Science Foundation (200020_182668,
250320_163243 and 206021_170729) and the Natural Sciences and Engineering Research Council of Canada
(NSERC CGSD3-547016-2020).

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
