# Peer review of "Opinion: Eliminating aircraft soot emissions"

_Aerosol Research, 2023_

## Author Comment (AC1)

We would like to thank Dr. Stettler for taking the time to thoroughly read our submission and offer his thoughtful *comments* that help us to bring forward our opinion even more clearly. We hope our itemized replies below and proposed text changes address his concerns but if anything remains, we would welcome any input.

*This opinion paper aggregates literature in a useful way, however there are several gaps that need addressing, detailed below. Furthermore, it is not entirely clear what the opinion is - is this paper supposed to put forward a view that soot emissions should be minimised? It currently appears that this is more of a review article.*

Indeed, after first reviewing in some detail the state of the art in the field, we expressed our opinion that realistic and science-based descriptions of aircraft soot emissions are needed highlighting thus opportunities for contributions by aerosol scientists. For example, even in our abstract we stressed the issues stemming from oversimplification of jet fuel soot characteristics (lines 11-13). At its end, we stress that existing technologies for reducing jet fuel soot emissions through combustor and fuel design are reviewed to identify strategies that eliminate aircraft soot emissions (lines 19-20). Isn't this a loud and clear opinion? Nevertheless, in the revised version of our manuscript we will stress further that realistic descriptions of jet fuel soot structure and composition are needed rather than the simplistic ones in today's otherwise very sophisticated models of fluid and energy dynamics in the operation of jet turbines for elimination of their emissions. Such realistic descriptions have, for example, been used quite effectively to describe black carbon (BC) formation and growth from a variety of combustion sources and even facilitate monitoring of BC emissions by aerosol (e.g., particle mobility and mass analyzers), laser (e.g., light extinction) diagnostics and fire detectors by accounting for BC morphology and limiting the current uncertainty regarding BC mass and particle size. In addition, by capitalizing on the accurate description of the high temperature residence time during enclosed combustion synthesis of nanomaterials and the latest advances in soot structure and composition, more than 99% of the emitted soot mass and concentration from enclosed jet fuel combustion was removed. Even though something like this had been stated explicitly in our p. 7, lines 327 – 329, this will be expanded as above. Similarly, a realistic description of BC allowed for the first time to determine conditions for synthesis of carbon black (CB) with closely controlled structure and size that is crucial for its diverse applications where for tire reinforcement hard agglomerates consisting of large primary particles (PP) are needed as fillers while for battery electrodes such agglomerates should consist of much finer PP and for inks or paints the CB agglomerates should be soft ones. Clearly such an understanding should be incorporated into the design of aircraft engines burning fossil and/or sustainable aviation fuels as it greatly facilitates engine design and operation for complete oxidation of any soot formed before its emission. This point will be emphasized in our abstract, text and conclusions in the revised paper version.

*The comments below represent significant omissions and I do not recommend publication of this article in it's current form.*

*Major comments:*

*1a. The authors have used the ICAO emissions databank to show data on nvPM emissions indices for different combustor types. It would have been useful to show this as a trend in time in addition to with respect to engine rated thrust.*

We agree that a trend with time would be useful, however, the only 'time' that is given in the ICAO database are the initial and final test dates. We have plotted this in Fig. R1 below showing the nvPM number (#/kg) emissions at a) idle and b) take-off. There appears to be no trend with the initial test

date. This is likely because it does not account for the production or design date of the engines so older engines could have been tested at a later date.

[Figure]

Figure R1: The nvPM number as a function of the initial test date at (a) idle/taxi (7% thrust), (b) take-off (100%). Combustor types represented in the database include SAC (squares), DAC (diamonds), RQL (circles), LDI (inverted triangles), LPP (triangles). The total nvPM number is normalized by the fuel flow (kg).

*1b. Furthermore, a discussion of the nvPM mass could also be shown.*

The nvPM mass is also important and it is shown in Fig. R2 below with the nvPM mass (mg/kg) at a) idle and b) take-off. As we had stated in the paper (p. 9 lines 409 – 410), the nvPM mass shows a similar trend to the nvPM number. The main differences between nvPM mass and nvPM number are the LPP values falling closer to the other combustors with mass-based emissions compared to number based emissions. This suggests that the LPP produces fewer but larger particles than other combustors on average. This will be stated in the revised paper on the current page 9 around line 410.

[Figure]

Figure R2: The nvPM number as a function of the rated thrust at (a) idle/taxi (7% thrust), (b) take-off (100%). Combustor types represented in the database include SAC (squares), DAC (diamonds), RQL (circles), LDI (inverted triangles), LPP (triangles). The total nvPM mass (mg) is normalized by the fuel flow (kg).

*1c. There is no mention on whether this data shown has been corrected for line-losses. Discussion on suggestions on improving or adding to the regulatory measurement procedure would be a welcome addition.*

Data submitted to the ICAO database should be collected following the procedure outlined in the ICAO Annex 16, Vol. II (ICAO, 2017). Briefly, particles are sampled at the engine exhaust with a no more than 35 m long (from probe tip to instrument inlet) heated sampling line to the measurement devices. This relatively long line, paired with the small size of aircraft soot may result in significant diffusional and thermophoretic losses due to temperature gradients as the sample cools from the exhaust temperature to sample line temperature. Since 2017, the nvPM mass and number diffusion and thermophoretic losses must be accounted for with the methods outlined in the ICAO Annex 16, Vol. II (ICAO, 2017). However, it is important to note that these losses are size-dependent, but the regulations do not require particle size measurements. Therefore, the estimate of the line loss correction may not be accurate for all engines. This will be stated in the revised version of our paper in current pg. 9, around line 407.

*1d. Discussion on how ground-level measurements scale to cruise conditions would also be welcome, e.g. https://egusphere.copernicus.org/preprints/2023/egusphere-2023-724/*

We had stated (pg. 2, line 50): As these emissions are measured only at ground level for the LTO cycle, the emissions most relevant for climate considerations are only indirectly estimated (Stettler et al., 2013). Subsequent to this sentence in the revised version of our paper, we will add: Estimates of emissions inventories must convert values measured at the ground to account for the drastically different atmospheric conditions at cruise (Teoh et al., 2023 Preprint).

*2. There is extremely limited discussion on the role of other aerosol particles in contrail formation. This is literature going back more than a couple of decades looking at the effect of sulphur, and there is emerging evidence that lubrication oil particles might play a role in the case of low soot conditions (https://egusphere.copernicus.org/preprints/2023/egusphere-2023-1264/). Consideration should be given to the potential contrail impacts under low soot conditions (https://www.nature.com/articles/s41467-018-04068-0).*

Of course, other aerosols may affect contrail formation as discussed in the cited pertinent literature (i.e. Kärcher, 2018). This is an active field of research especially in low soot conditions. As our paper is on eliminating aircraft soot emissions rather than on contrail formation, we only report the current understanding on the role of soot in contrail formation. Nonetheless, in the revised paper we will state around current line 60 (p. 2) that the role of soot in contrail formation is still unclear.

*3. There is no mention of aerosol cloud interactions. These is the most uncertain contribution of aviation to climate change and might be the largest contribution to RF, however both the sign and magnitude the RF is extremely uncertain (https://www.sciencedirect.com/science/article/pii/S1352231020305689). Emerging evidence suggests that the role of soot particles might be less important than ambient particles (https://egusphere.copernicus.org/preprints/2023/egusphere-2023-2441/; https://agupubs.onlinelibrary.wiley.com/doi/full/10.1029/2022JD037881). It is critical that this is covered in the article.*

We had already mentioned aerosol cloud interactions on pg. 1, line 37 "Soot emissions can impact the climate by warming the atmosphere through direct Radiative Forcing (RF) and indirectly by altering cloud processes and decreasing snow albedo (Bond et al., 2013)." Further detail was not given as this is an active area of research with large uncertainties that deviates from the topic of eliminating aircraft soot emissions. To highlight this, we will state in the revised version of our paper on pg. 2, around line

60 that there is high uncertainty in the RF of aviation aerosol-cloud interactions (i.e. indirect RF) and therefore no best estimate is given by Lee et al. (2021).

As with contrail formation, aerosol-cloud interactions is an active field of research deserving an opinion paper. Perhaps Prof. Stettler might be interested in contributing one highlighting opportunities for aerosol scientists as we tried to do here for eliminating soot emissions in which we had some first-hand experience in soot formation, growth, oxidation and interaction with light over the last 40 years, in contrast to aerosol-cloud interaction or contrail formation where we are eagerly looking forward to their experts' opinions.

**References:**

Teoh, R., Engberg, Z., Shapiro, M., Dray, L., and Stettler, M. E. J.: A high-resolution Global Aviation emissions Inventory based on ADS-B (GAIA) for 2019-2021, 2018, 1–27, 2023. Preprint.

---

## Author Comment (AC2)

**Aerosol Research**, ar-2023-15: Opinion: Eliminating aircraft soot emissions by Trivanovic & Pratsinis

**Reviewer 1:**

*General review*
*The authors present an opinion paper on "eliminating aircraft soot emissions"*
*According to the abstract the authors aims to give a review of existing technologies to identify strategies to eliminate soot emissions. From my point of view, they have not succeeded to accomplish this aim. There are some important technologies missing from the article, and the actual technologies leading the soot emission reduction are only mentioned briefly and for some reason in the description seems that these technologies are somehow still on a development phase.*

We thank the reviewer for the *comments* and hope our replies and text changes have fully addressed them but if anything remains, we would welcome any input.

*Overall the structure is correct but, in some points there are some repetitions or text that seems to be out of place. For example, in introduction, the last paragraph seems quite out of place.*

The last paragraph of the introduction defines, so-to-speak, the boundaries of this paper as "emissions from aviation" is a huge topic and we need to alert our readers to our focus.

*Detailied review:*

*Introduction*

*1. Lines 22-25, you state that aviation has a significant impact on health and climate due mainly to soot, but then you give two citations that deal with soot in general. Indeed, the review from Niranjan and Thakur mainly compile works on synthetic soot like printex and some on diesel soot, but not aeronautic soot. There is not too many works dealing with health effects of aircraft emissions, but still there are some that are much more relevant for the context than the review paper cited (for example: Bendtsen et al. 2019 https://doi.org/10.1186/s12989-019-0305-5, Delaval et al. 2022 https://doi.org/10.1016/j.envpol.2022.119521)*

The review by Niranjan & Thakur (2017) was chosen as it describes the biological mechanisms associated with exposure to combustion-generated carbonaceous aerosols like diesel soot and carbon black. As such, it is cited most appropriately. We thank the reviewer and inserted on page 1, line 26 "…the adverse health effects of aircraft emissions are partly due to non-volatile particle matter (Delaval et al., 2022) and aircraft soot has similar toxicity to diesel exhaust particles (Bendtsen et al., 2019)."

*2. Lines 27-29 you are mistaking jet aircraft smoke visibility and smoke number. Both can be linked (Slusher 1971 FAA report FAA-RD-71-23) but the smoke number is defined as the loss of reflectance of a filter used to trap smoke particles from a prescribed mass of exhaust per unit area of filter (SAE ARP 1179).*

To the best of our knowledge, the smoke number (SN) was the regulatory measure for particulate matter from aircraft that preceded the current non-volatile particulate matter (nvPM) standard. The purpose of the SN was to quantify and control visible pollution from aircrafts (Aircraft engine emissions, 2023) as is frequently stated in the literature (i.e. Stettler et al., 2013). To clarify this, we have now stated explicitly on page 1, line 29 that the smoke number was intended to control visible smoke from aircraft engines.

*3. Lines 64-65, why do you give a citation about light-duty diesel engine to illustrate soot NOx trade off in aircraft engines?*

The paper by Kim et al. (2009) was chosen as it neatly illustrates the relationship between nitrogen oxides (NOx), soot, carbon monoxide (CO) and unburned hydrocarbons (UHC) as a function of temperature. The correlation with temperature is important as it is what drives the increase in NOx and is applicable to all combustion with nitrogen present (i.e. combustion with air). The literature on soot-NOx trade-offs is less thorough but does exist. To highlight this, we now state that on page 2, line 74 that "…there is a well-established trade-off between soot and NOx in diesel engines (Kim et al., 2009) which has been observed also with aircraft combustors (Harper et al., 2022).

*4. Lines 67-69, I would like to see a reference about the uncertainties of soot contribution to climate in global models being due to the simplification of soot morphology*

The effect of this simplification is described in detail by Kelesidis et al. (2022) cited previously on lines 63 and 70 and now again on line 79. For completeness, we insert "…and composition" on page 2, line 80 because of the uncertainty in soot refractive index.

*5. Lines 92-99, you may want to have a look to this paper in ACPD https://doi.org/10.5194/egusphere-2023-2441*

Thank you for the suggestion and please note that this preprint was submitted after our own manuscript so it was not cited but it is an important step towards filling the gap in the literature that we mention here. We now add on page 3, line 107 that "…although recently there have been efforts to assess the ICN activity of soot from modern in-use commercial engines (Testa et al., 2024)".

*6. Lines 107-110 the definition of nvPM should have ben given at the beginning of the introduction*

The definition has been moved to page 1, line 34.

*7. Line 111, the discussion about >26.7 KN engines could be moved to line 32*

This discussion has been moved to page 1, line 37.

*8. Lines 116-118, what Avgas have to do with soot? Lead is a contaminant strictly linked to the fuel, I do not see the link with soot reduction or combustor technologies.*

Avgas is responsible for lead-containing aerosols internally or externally mixed with soot now stated on page 3, line 128. As such, it is an important source of lead pollution and an issue which many are unaware of. As in this paragraph we are highlighting the important issues that are outside the scope of this manuscript, we have modified the sentence as just above.

*Section 3.1*

*9. Frist of all, be careful with the term alternative fuel, not all alternative fuels are SAF.*

Indeed, we have chosen to use the phrase alternative fuels to make a more broad statement and include fuels not limited to SAF. To make this more clear, we now explain on page 6, line 253 that alternative aviation fuels include any fuels aside from kerosene-based jet fuels and Avgas. This includes, for example, hydrogen, ammonia and jet fuels made without fossil fuels. Sustainable aviation fuels (SAF) are non-fossil fuel jet fuels that are attractive due to…

*10. I am surprised by not seeing any references to CORSIA criteria when speaking about SAF.*

We have now clarified that the ICAO, under the Carbon Offsetting and Reduction Scheme for International Aviation (CORSIA), certifies alternative fuels as SAF based on a standardized Life Cycle Analysis (LCA) on page 6, line 271.

*11. Lines 232-233 No, alternative fuels can not be used as drop-in fuels. They can only be used after blending with jet fuel up to the mixing ratios defined by ASTM for each fuel. There are two alternative fuels which have started the procedure to be certified for their use as 100% drop-in fuel, CHJ and FT-SPK-A but they are not certified yet.*

We slightly disagree. Alternative jet fuels can be considered 'drop-in' when they do not require new fuel systems, distribution networks or new aircraft (ICAO, 2018), as now stated on page 6, line 258. SAF meets these definitions with goals to certify pure SAF in the future as stated by the reviewer. To ensure this is clear we now state on page 6, line 264 that …Currently, SAF are only certified for use when blended with conventional jet fuel although efforts are being made to certify 100% SAF in the future. Furthermore, we point out that SAF are attractive due to their potential to act as a drop-in solution for reducing $CO_2$ emissions on page 6, line 255.

*12. 238-239 No, LCAF are defined as: A fossil-based aviation fuel that meets the CORSIA Sustainability Criteria under this Volume (ICAO Annex 16 Vol IV), so indeed LCAF are fossil fuels that manage to reduce their lifecycle emissions by 10% compared to the baseline of 89 gCO2e/MJ*

We agree and removed the discussion of LCAF.

*13.  Line 258 that is the actual blend limit for HEFA not for SAF, other SAF are certified for much lower blend ratio, for example SIP is certified just up to 10% blend ratio.*

On line 284 we state that 50% is the current upper limit for a SAF blend, we do not say that all SAF blends are certified to this blend ratio. So, no change has been made.

*14.  Line 290, I am missing references to on-going project like ECLIF, EcoDemonstrator or VOLCAN, where in flight measurements has been done even with 100% SAF*

Indeed, since 2021 several 100% SAF flights have been made with measurements at cruise altitudes. However, to the best of our knowledge, the results of these campaigns had not yet been published at the time our manuscript was submitted. Since our submission, one paper describing the results of the above mentioned ECLIF3 project has come online showing an apparent decrease in ice crystal number. We now state on page 6, line 265 "Flights powered with 100% SAF have already been performed for research purposes (Märkl et al., 2024)"

*15.  Lines 304-307, Fit for 55 and ReFuelEU aviation target for a share of SAF in the annual fuel used, not in the blends.*

We agree. To ensure this distinction is clear, we have now stated on page 7, line 336 that the European Commission and the US have implemented policies to mandate the annual uptake of SAF… We cannot change the wording of "all aviation fuel supplied to aircraft operators at (European) Union airports contains a minimum share of SAF" (European Commission, 2021) as this is a direct quote.

*16.  Indeed, how things work actually is just as the author describes, the blends used in airports are normally blends of around 30% SAF. It also has to be noted that even if the EU fix a minimum amount of SAF share by year, industry have proposed higher SAF shares, for example Airbus have set as an objective to use 10% SAF share for 2023 for internal movements and 5% for external movements,*

While companies can place goals for SAF use, it might not be possible to meet such goals due to the limited supply. The International Air Transport Association (IATA) pointed out that "despite a significant price difference between conventional jet fuel and SAF, every single drop of sustainable aviation fuel produced was purchased by aircraft operators and their customers." (IATA, 2023). Nonetheless, in 2022 SAF was only 0.1 to 0.15% of total jet fuel demand.

*17. and again this is done by using blends between 30 and 49% of SAF for some flights, and not blends 1% of SAF for all flights. So last sentence of the paragraph (306-307) are incorrect and must be removed.*

The purpose of our statement in these lines is to highlight the missed opportunity when spreading a limited supply across many flights compared to targeted use on specific flights. We have not stated that 1% blends *will* be used. While there are operators using the highest allowable blends, some mandate the use of SAF on *all* flights or are vague regarding the blend percentage. For example, Air France KLM promises they "will add a percentage (0.5% to 1%)

of SAF on all flights departing from France and the Netherlands" (Air France KLM Sustainable Aviation Fuel) now stated on page 7, line 340. Regarding the ReFuelEU regulations, the EASA has stated: "This means that every flight leaving the larger EU airports, will carry a minimum amount of SAF" (Fit for 55 and ReFuelEU Aviation), now quoted on page 7, line 339. These policies come into effect in 2025 and current practices may not be influenced by these policies. For these reasons, no change has been made.

*18. After reading SAF section I have the feeling that the authors are not familiar with aeronautic SAF. There are too many wrong or misleading information.*

We hope that our replies above and subsequent changes to the text have addressed the reviewer's concerns but if anything remains, we would welcome any input.

*Section 3.2*

*19. Overall all this section is too naïf to be a review of existing technology, only speak of general combustor concepts, but there is no mention of different technologies, just as an example, there are different RQL implementations (LEC, TALON ...) and each of this have their own specificities.*

We agree that there are many different technologies not described here. However, due to restrictions on article length we could not describe all aircraft engines and chose instead to focus on the general combustor concepts which are most relevant today to the audience of this journal, largely aerosol scientists and practitioners. Nevertheless, we now state on page 10, line 463 "Within these broad categories, there are multiple distinct implementations of these combustor types. For example, the RQL category contains the Rolls-Royce Phase5 series, Pratt & Whitney Talon series and General Electric LEC series combustors" to alter the readers to the different implementations.

*20. I am missing also TAPS.*

We did not mention TAPS specifically as it is a single implementation of a more general combustor concept, in this case the lean premixed prevaporized concept.

*21. Regarding technologies in development, they only mention LEAF, while there are other technologies that are much more advance in development, for example WET engines that will be tested on flight in the near future.*

While there are indeed other technologies in development, this manuscript is discussing knowledge on the soot aerosols emitted today. The Water-Enhanced Turbofan (WET) concept expects to significantly reduce soot emissions during the stage where water is captured for recirculation (Kaiser et al., 2022). However, to the best of our knowledge, there are no scientific studies published on the actual measured or modeled soot emissions from WET engines now stated on page 11, line 500.

*22. Also developments like open-rotor engines might be included, thought it is not directly linked to combustor optimization, but again this technologies will be tested on flight soon*

*while LEAF is at laboratory level demonstration.*

We now explain on page 7, line 350 that …some in-development technologies, such as open rotor engines, promise significant reductions in fuel consumption (Khalid et al., 2013) and soot emissions but are not discussed here as they are not linked to the actual formation of soot. Due to length restrictions, we can only cover the topics which are most relevant to aerosol scientists and practitioners.

*23. Lines 327-329 results shown in Kelesidis et al. 2023 has been obtained with a laboratory combustor, that has nothing to do with a real aircraft combustor, despite they might obtain soot with similar properties that soot obtained in some engine regimes, results about oxygen injection cannot be transferred to a real engine.*

To clarify this we now emphasize on page 8, line 367 that "…it was shown in a laboratory setting that a judicious injection…" and on page 8 line 369 "Such results can be transferred to real engines by matching the high temperature residence time as has been shown with scale-up of flame synthesis of nanoparticles from mg to kg per hour (Kelesidis and Pratsinis, 2021)."

*24. Line 408 Emission index is defined as emissions by kg of burned fuel.*

This is probably a misunderstanding. We have provided the nvPM emissions per kg of fuel as given in the ICAO database.

*25. Lines 412-414, I wondering how the authors have assigned different engines to different technologies. I do not see where they have obtained LDI engine data, or LPP, are they assuming that TAPS is equivalent to LPP? according with line 416 you include in your graph 27 LPP engines, in the data base there is data for 26 TAPS and TAPS II engines, so seems so (though I don't know what is the extra engine you consider LPP to sum up the 27 points you use),*

Yes, for the purposes of comparison, TAPS has been considered LPP. The reviewer is correct in pointing out that it is 26 engines not 27 and this has been changed in the text on page 10, line 463.

*26. Furthermore, what is the point of comparing old SAC engines with new SAC-TI engines?*

All of the data available on the ICAO website has been used to understand the breadth of the existing knowledge. As it is not possible to go into detail on every implementation in one brief article, the distinction has not been made.

*27. The authors claim that RQL combustors have a large variation, and that this is most likely due there is more data entries for RQL… no, this is due to the fact that you are comparing technologies based on RQL concept but that are differently implemented and also that you are probably including out of production engines (ICAO data base contain data for*

*215 engines, 43 of those are out of production, your graph includes 208 points).*

We agree with the reviewer that the RQL section includes a variety of technologies and this is the reason for the variation and this is what we had intended to say as the data entries are not just repeated measurements of the same engine. We now explicitly state on page 10, line 469 that RQL combustors have the most variation …likely due to the variety of different implementations of the RQL concept…

*28.  Line 440-441 As this is written I interpret that you say that engine manufacturers manipulate the data included in ICAO data base, what is unacceptable , this is just a plain offense to ICAO, SAE-E31 and EASA.*

Of course, we do not intend to imply this and are surprised to read this. By the way, for a small number of engines, research measurements are also included (ICAO Aircraft Engine Emissions Databank, 2023) now stated on page 11, line 491.

*29.  Line 446-448 again authors are forgetting a large number of on-going projects linked to these engines. The end of this sentence " … if they will be adopted in the future"  just illustrate the lack of understanding of the authors in the aeronautic field. LEAP and GEnx engine has been in the market for several years, actually there are almost 10k on service, and if you check the orders of engines only for LEAP engines, this indicates an annual production for the next year of over 2000 engine per year.*

We kindly disagree with the reviewer. We have not said that technologies will not or have not been adopted but rather that …There is relatively little scientific research on soot emissions from novel aircraft engines… (page 11, line 499) and once these technologies are in service, they have a long service life.

*30.  Line 450-452 Again an other sentence that indicates that the authors are not familiar with the engine development. CFD combustion models are used in the development of engines, but indeed they represent only a small part on the development, there are several test of different injector configurations, combustion chamber geometries etc, so I do not see the point of the authors here.*

CFD is just one example we gave of how low emission technologies could be identified before significant investment is put into empirical testing of the best design. Our aim is not to describe how manufacturers develop engines, but rather to identify areas such as the fundamental understanding of soot formation and elimination, where aerosol scientists can contribute meaningfully.

*Conclusion*

*31.  I have the feeling that authors are neglecting the potential of engines based on TAPS combustor to reduce both NOx and soot emissions. Indeed, above 30% of engine thrust soot emissions of these engines are close to be under the detection limit of the instruments.*

As stated above, while we have not explicitly discussed the TAPS implementation, we do state that this category of combustors performs the best of all the combustors reported in the ICAO database. To make this more clear we now state on page 10, line 472 that all of the LPP combustors are from the TAPS combustor series.

*32.  The main conclusion seem to be that the solution for eliminate soot from aircraft emissions is the development of CFD models able to predict soot production. This will be of course useful, but I do not think is something realistic. Further more actual models used in engine development are able to give at least order of magnitude values for soot production, and in any case all engine development goes through an experimental optimization of injectors and combustion chamber.*

We kindly disagree with the reviewer. It is not realistic to predict soot formation without properly accounting for its shape (Bouaniche et al., 2020). For example, models overpredict soot volume fraction by up to 3 times when particles are assumed to be spherical (Kelesidis and Pratsinis, 2021), now stated on page 5 line 241. In fact, CFD is only mentioned once in the entire article and not in the conclusion! As this has been submitted to an aerosol journal, our aim is to give aerosol scientists an overview of the problem and an understanding stated on page 12, line 518 that without a robust understanding of the role of soot in direct and indirect radiative forcing (e.g. through contrail formation) trade-offs between soot reduction and other pollutants cannot be properly accounted for. CFD is just one of the tools that can be used to better understand soot formation and elimination from aircraft emissions.

*33.  I do not see that this article bring something new to the field, even as a review there are many things missing. For me the question right now in the field  is not so much to reduce further soot emission from actual engines ( at least for thrust > 30%) but, what could be other sources of particulate matter once the soot emissions are reduced at ambient background levels and how contrail formation can be affected. Other interesting topic can be if is worthy to produce Lean burn engines or SAF + rich burn engines can achive  similar reductions on nvPM emissions.*

We kindly disagree with the reviewer. Our aim is to highlight the state of knowledge and the places where, aerosol scientists can contribute most effectively to aerosol formation and elimination from aircraft emissions.

**Reviewer 2:**

*This opinion paper addresses the issue of soot emissions from aircraft engines, which is a significant contributor to air pollution and climate change. The authors review current strategies and technologies for reducing soot emissions, including combustor designs, the use of alternative fuels (SAF), and discussing trade-offs with NOx. They also highlight the challenges in accurately modelling and measuring soot emissions, which complicates efforts to design effective emission reduction strategies.*

*The paper is well-written with good discussions on soot formation and aircraft combustors. It is of scientific interest and makes some interesting points particularly regarding the targeted approach for SAF usage. I have a few minor comments and suggestions for the authors to consider:*

We thank the reviewer for the *comments* and hope our replies and text changes have fully addressed them but if anything remains, we would welcome any input.

*Main comments:*

1. *While the perceived benefits from SAF and of a targeted approach for its implementation are discussed in detail, I'd like to point out that because SAF can currently only be blended at up to 50%, the soot emissions from the SAF blend are ultimately driven by the jet-A it's being blended with. By that, I mean that the soot emissions from a relatively clean jet-A (fuel hydrogen content > 14.2% for example) might be lower than that of a dirty jet-A (fuel hydrogen content <13.2% for example) blended with SAF because ultimately, it's the fuel hydrogen content that correlates with the soot emissions. I thought this could be something you could discuss in your paper.*

   We agree that the composition of the conventional jet fuel that is blended with SAF is important for assessing the final emissions. Therefore, we have added on page 6, line 296 that as the aromatic content and H/C of conventional jet fuels varies, the actual reductions achieved with SAF blends will depend on composition of the conventional jet fuel, particularly when the SAF blending ratio is low.

2. *While the main topic of your paper is soot emissions, there is no mention of other PM emissions from aircraft engines such as lubrication oil, volatile PM, secondary organic aerosol (SOA) which could also significantly contribute to air pollution and climate change by either coating soot or forming new particles in the plume. Some of these emissions, such as sulphates particles, are dependent on fuel properties and therefore could be reduced with SAF. I suggest discussing this as well in your manuscript.*

   We agree as well that non-soot PM emissions are also important components of aircraft emissions. However, due to article length constraints it is not possible to cover all of these PM emissions. We have focused on the non-volatile particles (primarily soot) which are currently regulated by the ICAO while volatile PM is not regulated. We have added to pg. 1 line 37 that volatile particles, lubrication oil particles and secondary organic aerosol may also have important health and climate impacts however, they are not currently regulated and so will not be covered here.

3. *While you introduced and discussed EC/OC from aircraft soot, I didn't understand from your manuscript how this parameter was relevant to health and the environment. Why is EC/OC important in this context (e.g., tracks combustion efficiency, EC warming effect, OC contribute to SOA)?*

   Indeed, we had stated already on the original page 2, line 76 "…quantifying OC/EC ratios is important for understanding the light absorption of soot (Kelesidis et al., 2021)…" and current line 141 " The OC/TC influences the optical properties of soot and thus its RF (Kelesidis et al., 2021)." To further expand on the importance of the OC/EC ratios we now state on page 3, line 95: "The OC/EC ratio is important for source apportionment of ambient aerosols (Ramadan et al., 2000), attempting to understand the health effects of soot (Kelly and Fussell, 2012) and for determining the light absorption of soot (Kelesidis et al., 2021)." and on Line 141: "…and will impact the output of optical instruments used

to measure aircraft emissions (Durdina et al., 2016)."

**Minor comments:**

4. *Line 3: You define "Nitrogen oxides (NOx)" but not "CO2" (carbon dioxide) – I suggest being consistent.*

   Done.

5. *Line 8: the term "mobility" is not introduced or discussed in the manuscript. You actually refer here to "electrical mobility" (as measured by SMPSs, DMS500, EEPS, etc) which is an equivalent diameter and I believe this should be discussed, particularly in regards with modelling, which maybe considers a different particle equivalent diameter (e.g., aerodynamic diameter) which could contribute to uncertainty.*

   We have now elaborated on pg. 3 line 99 "The size of irregular agglomerates such as soot is quantified by equivalent diameters for example, electrical mobility diameter, $d_m$ (Fig. 1: broken line), aerodynamic diameter or projected area equivalent diameter where the type of equivalent diameter depends on the measuring principle. Such agglomerate diameters can several times larger than the mass-equivalent diameter typically calculated by models (Eggersdorfer and Pratsinis, 2014) that could contribute to uncertainty"

6. *Line 9: what do you mean by "quite high" – please quantify or use the word "relatively" instead.*

   To be quantitative we now state that "…at low thrust the OC/TC can be quite high (> 75%)" on page 1, line 9.

7. *Line 72: typo (return to line).*

   Done.

8. *Line 81 (figure title): "broken line": Do you mean red dashed line? Also, primary particles are solid black lines, not red.*

   Indeed, we are referring to the dashed (or broken) line. The solid line refers to the exemplary primary particle highlighted on the far right of the schematic. To avoid confusion, we dropped the word "red" from the caption as all lines are red.

9. *Line 128: Can you discuss in more details how OC/TC influences the environment (see comment above).*

As stated in our response to comment 3, *OC/TC* is important for understanding the climate impact of soot as it determines the radiative forcing of soot (Kelesidis et al., 2022) and will impact the output of optical instruments used to measure aircraft emissions (Durdina et al., 2016) now stated on pg. 3, line 141.

10. *$d_p$: To my knowledge this abbreviation typically refers to particle diameter and not primary particle size, I suggest replacing it by $d_{pps}$ or $d_{pp}$ throughout the manuscript to prevent confusion.*

While many use $d_p$ to refer to primary particle diameter (Kholghy et al., (2013), Camacho et al., (2015) and Saffaripour et al., (2017) to name a few) it is true that many also use $d_{pp}$. So, to avoid any confusion we have replaced $d_p$ with $d_{pp}$.

11. *L194: replace "poor" by "poorer".*

Done.

12. *L255, 279: Other more recent scientific literature has correlated soot emissions with fuel hydrogen content (e.g., https://doi.org/10.1016/j.fuel.2020.119637, https://doi.org/10.1016/j.fuel.2021.123045, https://doi.org/10.1016/j.fuel.2019.115903), I suggest adding them where relevant.*

We thank the reviewer for the suggestions. We have now stated on pg. 6, line 286 "In fact, the geometric mean $d_m$ has been shown to drop nearly linearly as H/C increases while decreases in nvPM number were not as steep suggesting that the decrease observed in nvPM mass is strongly influenced by the smaller particle sizes for HEFA-based fuels (Durand et al., 2021). Similar trends were observed for a range of different SAF types including HEFA, Alcohol-to-Jet (ATJ) and a Catalytic Hydrothermal Conversion Jet (CHCJ) fuel showing that the dependence on H/C is not dependant on the fuel production method (Harper et al., 2022)." and on pg 7. Line 304 "However, if a SAF blend is designed with higher aromatic content and lower H/C than a conventional fuel, soot emissions could even increase (Schripp et al., 2019)."

13. *L466: Explain why EC/OC matters.*

In addition to our replies to comments 3 and 9 we have added the following statement on page 12 line 523: "The OC/TC ratio of aircraft soot, which has implications for the source apportionment (Ramadan et al., 2000), health effects (Kelly and Fussell, 2012) and optical properties of soot (Kelesidis et al., 2021), depends on thrust (Elser et al., 2019; Fig. 6)."

14. *L474/475: this sentence says that soot emissions are reduced with lower H/C content when its reality it's the other way around (higher H/C ratios result in lower soot emissions). Please rephrase.*

We agree and have reformulated the sentence on page 12, line 540 to state that "Sustainable Aviation Fuels (SAF) have the potential to significantly reduce soot

[revised manuscript text omitted]